# LANGUAGE MODELING IS COMPRESSION

**Grégoire Delétang**[*1]     **Anian Ruoss**[*1]     **Paul-Ambroise Duquenne**[2]     **Elliot Catt**[1]

**Tim Genewein**[1]     **Christopher Mattern**[1]     **Jordi Grau-Moya**[1]     **Li Kevin Wenliang**[1]

**Matthew Aitchison**[1]     **Laurent Orseau**[1]     **Marcus Hutter**[1]     **Joel Veness**[1]

## ABSTRACT

It has long been established that predictive models can be transformed into lossless compressors and vice versa. Incidentally, in recent years, the machine learning community has focused on training increasingly large and powerful self-supervised (language) models. Since these large language models exhibit impressive predictive capabilities, they are well-positioned to be strong compressors. In this work, we advocate for viewing the prediction problem through the lens of compression and evaluate the compression capabilities of large (foundation) models. We show that large language models are powerful general-purpose predictors and that the compression viewpoint provides novel insights into scaling laws, tokenization, and in-context learning. For example, Chinchilla 70B, while trained primarily on text, compresses ImageNet patches to 43.4% and LibriSpeech samples to 16.4% of their raw size, beating domain-specific compressors like PNG (58.5%) or FLAC (30.3%), respectively. Finally, we show that the prediction-compression equivalence allows us to use any compressor (like gzip) to build a conditional generative model.

## 1 INTRODUCTION

Information theory and machine learning are inextricably linked and have even been referred to as "two sides of the same coin" (MacKay, 2003). One particularly elegant connection is the essential equivalence between probabilistic models of data and lossless compression. The source coding theorem (Shannon, 1948) is the fundamental theorem describing this idea, i.e., the expected message length in bits of an optimal entropy encoder is equal to the negative $\log_2$-likelihood of the statistical model. In other words, maximizing the $\log_2$-likelihood (of the data) is equivalent to minimizing the number of bits required per message. Indeed, lossless compression with a probabilistic model can be achieved in a variety of different ways, including Huffman coding (Huffman, 1952), arithmetic coding (Pasco, 1977; Rissanen, 1976), and asymmetric numeral systems (Duda, 2009).

Arithmetic coding, in particular, is known to be optimal in terms of coding length, meaning that the overall compression performance depends on the capabilities of the probabilistic model (see Fig. 1 for an overview of arithmetic coding). Incidentally, in recent years, large pre-trained Transformers (Vaswani et al., 2017), so-called *foundation models* (Bommasani et al., 2021), have proven to be highly successful across a wide range of predictive tasks (Bubeck et al., 2023; Rae et al., 2021) and are thus promising candidates for use with arithmetic coding. Indeed, Transformer-based compression with arithmetic coding has produced state-of-the-art results both in the online (Bellard, 2021; Mao et al., 2022) and offline settings (Valmeekam et al., 2023). In the online setting, a pseudo-randomly initialized model is directly trained on the stream of data that is to be compressed, while the offline setting, which we consider in our work, trains the model on an external dataset before employing it to compress a (potentially different) data stream. Consequently, offline compression is performed *in-context*, with a fixed set of model parameters. Transformers have demonstrated impressive in-context learning abilities (Laskin et al., 2023; Brown et al., 2020; Wei et al., 2022; Genewein et al., 2023) and are thus ideally suited for offline compression.

---

[*]Equal contribution. [1]Google DeepMind. [2]Meta AI & Inria. Correspondence to {gdelt, anianr}@google.com.

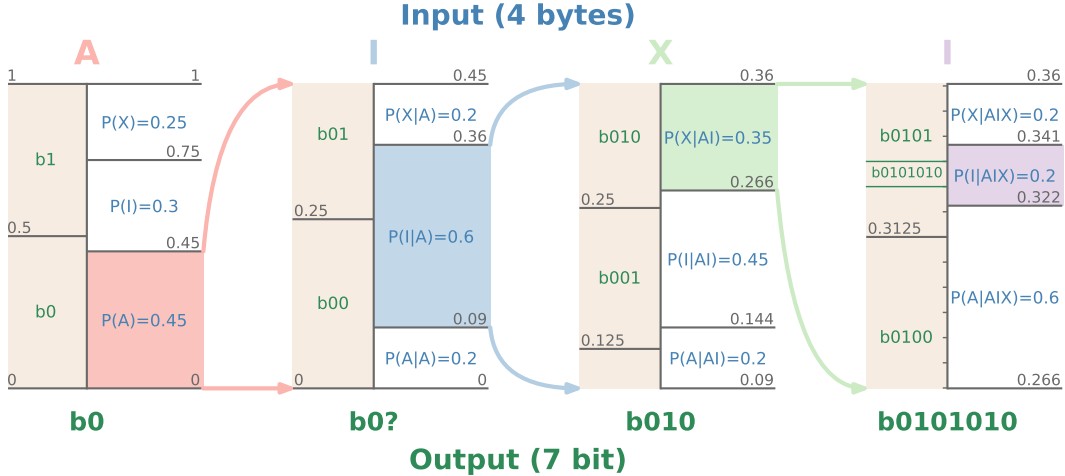

Figure 1: Arithmetic encoding of 'AIXI' with a probabilistic model $P$ (blue) resulting in the binary code 'b0101010' (green). We iteratively divide the real interval $I = [0, 1)$ according to the model's (conditional) probabilities and select the sub-interval corresponding to the observed symbol (e.g., $I = [0, 0.45)$ for $P(A)$). We further refine $I$ for each input symbol (indicated by the arrows), e.g., $I = [0.09, 0.36)$ for $P(I|A)$. To determine the encoded output, we iteratively split $[0, 1)$ in half and assign a binary code to each sub-interval (shaded red areas). At every step we can output the binary code if $I$ is fully contained in the corresponding binary interval (e.g., 'b0' for 'A', but not for 'AI' as it could be 'b00' or 'b01'). At the end of the input, the code is 'b0101', which cannot be uniquely decoded ($P(A|AIX)$, $P(I|AIX)$, $P(X|AIX)$ all overlap with 'b0101'). Thus, we further refine the binary code until its binary interval is fully contained in $I$ (all calculations in Appendix A).

The context length is a key limiting factor in offline compression, as it dictates the maximum number of bytes a model can compress at a time. Transformers can only compress a few kilobytes (each "token" being coded with 2 or 3 bytes), while requiring a lot of compute. Correspondingly, many challenging predictive tasks (e.g., algorithmic reasoning or long-term memory) require long contexts (Delétang et al., 2023), and thus extending these models' context lengths is a key challenge which is gaining increased attention (Zaheer et al., 2020; Guo et al., 2022; Bulatov et al., 2023). The in-context compression view provides insights into the failure modes of current foundation models.

**This Work** We advocate for using (lossless) compression to study foundation models. To that end, we conduct an extensive empirical investigation of the offline (in-context) compression capabilities of large language models, with the rationale that they have become readily available (Touvron et al., 2023a;b) and can thus be used for compression without the training overhead. We empirically demonstrate that these models, while (meta-)trained primarily on text, achieve competitive compression rates across different data modalities, outperforming domain-specific standard compressors (not accounting for model parameter size). Moreover, we shed new light on scaling laws (Kaplan et al., 2020), showing that they also hold true for compression but that measuring the adjusted compression rates instead of the log loss adds a twist: Scaling beyond a certain point will deteriorate the compression performance since the model parameters need to be accounted for in the compressed output. Finally, we advocate for framing (self-supervised) prediction through the lens of compression as it encompasses generalization: a model that compresses well generalizes well (Hutter, 2006).

**Contributions** We empirically study the lossless compression capabilities of foundation models:

- We review how to compress with predictive models via arithmetic coding and call attention to the connection between current language modeling research and compression.
- We show that large language models achieve impressive compression rates (disregarding model parameter size) on modalities other than text. For example, Chinchilla 70B achieves compression rates of 43.4% on ImageNet patches and 16.4% on LibriSpeech samples, beating domain-specific compressors like PNG (58.5%) or FLAC (30.3%), respectively.

- We revisit scaling laws, showing that the dataset size provides a hard limit on model size in terms of compression performance and that model scaling is not a silver bullet.

- We leverage the compression-prediction equivalence to employ compressors as generative models and visually illustrate the performance of the underlying compressor.

- We demonstrate that tokenization, which can be viewed as a pre-compression, does, in general, not improve compression performance, but allows models to increase the information content in their context and is thus generally employed to improve prediction performance.

## 2 BACKGROUND

In this section, we review the necessary background on information theory and its relation to likelihood maximization. To that end, we consider streams of data $x_{1:n} := x_1 x_2 \ldots x_n \in \mathcal{X}^n$ of length $n$ from a finite set of symbols $\mathcal{X}$. We write $x_{\leq j} = x_{<j+1} := x_{1:j}$ for $j \leq n$ and denote the empty string as $\epsilon$. Finally, we denote the concatenation of two strings $s$ and $r$ by $sr$.

**Coding Distributions** A coding distribution $\rho$ is a sequence of probability mass functions $\rho_n : \mathcal{X}^n \mapsto (0, 1]$, which for all $n \in \mathbb{N}$ satisfy the constraint that $\rho_n(x_{1:n}) = \sum_{y \in \mathcal{X}} \rho_{n+1}(x_{1:n}y)$ for all $x_{1:n} \in \mathcal{X}^n$, with the base case $\rho_0(\epsilon) := 1$. From here on out, whenever the meaning is clear from the argument to $\rho$, we drop the subscript on $\rho$. Under this definition, the conditional probability of a symbol $x_n$ given previous data $x_{<n}$ is defined as $\rho(x_n \mid x_{<n}) := \rho(x_{1:n})/\rho(x_{<n})$, with the familiar chain rules $\rho(x_{1:n}) = \prod_{i=1}^{n} \rho(x_i \mid x_{<i})$ and $\rho(x_{j:k} \mid x_{<j}) = \prod_{i=j}^{k} \rho(x_i \mid x_{<i})$ following.

**Lossless Compression** The goal of lossless compression is to encode a stream of symbols $x_{1:n}$ sampled from a coding distribution $\rho$ into a bitstream of minimal (expected) length, while ensuring that the original data sequence is recoverable from the bitstream. To that end, we use a binary source code $c : \mathcal{X}^* \mapsto \{0, 1\}^*$, which assigns to each possible data sequence $x_{1:n}$ a binary code word $c(x_{1:n})$ of length $\ell_c(x_{1:n})$ (in bits). Thus, the aim is to minimize the expected bits per sequence $L := E_{x \sim \rho}[\ell_c(x)]$, i.e., encoding rare sequences with more bits and frequent sequences with fewer bits. Shannon's source coding theorem establishes the limit on possible data compression as $L \geq H(\rho)$ for any possible code, where $H(\rho) := \mathbb{E}_{x \sim \rho}[-\log_2 \rho(x)]$ is the Shannon entropy (Shannon, 1948).

**Arithmetic Coding** Given a coding distribution $\rho$ and a sequence $x_{1:n}$, arithmetic coding (Pasco, 1977; Rissanen, 1976) constructs a code with almost optimal length. It directly connects coding and compression with prediction and modeling: compressing well means modeling well in a log-loss sense and vice-versa. Assuming infinite precision for the arithmetic operations involved, the arithmetic code has length $-\lceil \log \rho(x_{1:n}) \rceil + 1$ bits, whereas the optimal code length is $-\log \rho(x_{1:n})$ bits. A practical implementation that is subject to $B$ bit precision adds further $O(n2^{-B})$ bits (Howard & Vitter, 1991), which is negligible for 32- or 64-bit arithmetic. In the following we consider infinite precision arithmetic coders and refer to Witten et al. (1987) for the finite-precision implementation.

**Arithmetic Encoder** The arithmetic code of a sequence $x_{1:n}$ is the binary representation of a number $\lambda \in [0, 1)$. We identify $\lambda$ by narrowing down an interval that encloses $\lambda$ step by step (maintaining a growing prefix of the binary representation of $\lambda$ throughout the process). Initially, this interval is $I_0 = [0, 1)$. In step $k > 0$ (i.e., encoding $x_k$), we first partition the previous interval $I_{k-1} = [l_{k-1}, u_{k-1})$ into $N$ sub-intervals $\tilde{I}_k(x_1), \tilde{I}_k(x_2), \ldots$, one for each letter from $\mathcal{X} = \{x_1, x_2, \ldots, x_N\}$. The size of sub-interval $\tilde{I}_k(y)$ that represents letter $y$ is $(u_{k-1} - l_{k-1}) \cdot \rho(y \mid x_{<k})$. Formally, we define

$$\tilde{I}_k(x) := \left[ l_{k-1} + (u_{k-1} - l_{k-1}) \cdot \sum_{y < x} \rho(y \mid x_{<k}), \quad l_{k-1} + (u_{k-1} - l_{k-1}) \cdot \sum_{y \leq x} \rho(y \mid x_{<k}) \right), \tag{1}$$

assuming a strict order on $\mathcal{X}$. To encode $x_k$ we proceed with its corresponding interval, i.e., $I_k = \tilde{I}_k(x_k)$. Finally, we choose $\lambda \in I_n$ with the shortest binary representation in the terminating interval $I_n$ and use that binary representation to encode $x_{1:n}$. Fig. 1 illustrates this process.

**Arithmetic Decoder**   Given $\lambda$ and $\rho$ decoding the $k$-th letter is easy: Starting with $I_0 = [0, 1)$, find $y$ such that $\lambda \in \tilde{I}_k(y)$ to decode $x_k = y$, then set $I_k = \tilde{I}_k(x_k)$ and proceed with the $k+1$-st letter.

**Likelihood Maximization**   In practice, the source distribution $\rho$ is usually unknown and is instead estimated with a parametric probabilistic model $\hat{\rho}$. Thus, instead of achieving code length $-\sum_{i=1}^{n} \log_2 \rho(x_i \mid x_{<i})$ for the sequence $x_{1:n}$, we obtain the suboptimal length $-\sum_{i=1}^{n} \log_2 \hat{\rho}(x_i \mid x_{<i})$. As a result, the expected (suboptimal) number of bits is the *cross-entropy*:

$$H(\rho, \hat{\rho}) := \mathbb{E}_{x \sim \rho} \left[ \sum_{i=1}^{n} - \log_2 \hat{\rho}(x_i \mid x_{<i}) \right]. \tag{2}$$

Thus, we can minimize the expected length of the encoded data stream with symbols distributed according to $\rho$ by minimizing the cross-entropy with respect to some $\hat{\rho}$, which is equivalent to likelihood maximization (MacKay, 2003). However, Eq. (2) is exactly the same objective used to train current foundation models, i.e., the log-loss. Thus, minimizing the log-loss is equivalent to minimizing the compression rate of that model used as a lossless compressor with arithmetic coding, i.e., current language model training protocols use a maximum-compression objective.

**Compression-Based Sequence Prediction**   Analogous to how a predictive distribution can be used for lossless compression via arithmetic coding (described above), any compressor can be employed for sequence prediction (Frank et al., 2000). The main idea is to define $\rho(x_{1:n})$ as the coding distribution $2^{-\ell_c(\cdot)}$, where $\ell_c(x_{1:n})$ is the length of sequence $x_{1:n}$ when encoded with compressor $c$ (e.g., gzip). We thus recover the conditional distribution $\rho(x_i \mid x_{<i})$ by computing $2^{\ell_c(x_{<i}) - \ell_c(x_{<i}x_i)}$, for all $x_i$.

**Universal Coding**   Above we discussed optimal (arithmetic) coding with respect to data sampled from a fixed distribution $\rho$. In contrast, universal (optimal) source coding with respect to all computable sampling distributions can, in theory, be achieved by choosing $\ell_c(x_{1:n})$ as the Kolmogorov complexity of $x_{1:n}$ (Kolmogorov, 1998; Li & Vitányi, 2019). For this choice, the conditional distribution described above is universally optimal over $x_{<i}$, recovering the Solomonoff predictor (Solomonoff, 1964a;b; Rathmanner & Hutter, 2011). The Solomonoff predictor is a Bayesian mixture of *all* predictors that can be programmed in a chosen Turing-complete programming language. More precisely, for a predictor $q$ of program-length $\ell_c(q)$ bits, the Solomonoff predictor assigns a prior weight of $2^{-\ell_c(q)}$ to predictor $q$. That is, if $\mathcal{Q}$ is the set of all predictors that can be programmed and computed, the Solomonoff predictor assigns probability $S(x_{1:n}) = \sum_{q \in \mathcal{Q}} 2^{-\ell_c(q)} q(x_{1:n})$ to a sequence $x_{1:n}$. Therefore, $S(x_{1:n}) \geq 2^{-\ell_c(q)} q(x_{1:n})$ for all $q \in \mathcal{Q}$, and thus $-\log_2 S(x_{1:n}) \leq -\log_2 q(x_{1:n}) + \ell_c(q)$. Observe that $\ell_c(q)$ is a constant of $q$ that is independent of the sequence length. Therefore, compressing optimally is equivalent to predicting optimally and vice versa (Hutter, 2005).

## 3   EXPERIMENTAL EVALUATION

Here, we evaluate foundation models' (in-context) compression capabilities (details in Appendix B and code at `https://github.com/google-deepmind/language_modeling_is_compression`).

**Compressors**   We compare our arithmetic coding-based language model compressors to two competitive general-purpose lossless compressors: gzip (Deutsch, 1996) and its improvement LZMA2 (Pavlov, 2019), used by the 7zip software. Both are based on Huffman coding (Huffman, 1952) and the Lempel-Ziv-Welch algorithm (Welch, 1984). We also consider specialized lossless compressors for image and audio data, i.e., PNG (Boutell, 1997) and FLAC (Coalson, 2008), respectively. Finally, we evaluate two types of language models (of different sizes) with arithmetic coding: vanilla decoder-only Transformers (Vaswani et al., 2017), which we train on the enwik8 dataset, and the pretrained Llama 2 (Touvron et al., 2023b) and Chinchilla (Hoffmann et al., 2022).

### 3.1   DATASETS

We consider datasets of three different modalities, text, image, and audio, which have (a priori) very different biases for compression and thus provide a good testbed for evaluating a compressor's general capabilities. To render the results comparable across modalities, all our datasets are 1GB.

A key question is how to reconcile the different context lengths $C$ of the compressors we consider. Transformers are restricted to short contexts (2048 "tokens", coded over 1 byte for our trained transformers, and 4 bytes for the pretrained models), while gzip uses a maximum context of 32 kilobytes, and LZMA2 has a virtually "infinite" context length. Having a longer context allows a compressor to exploit more sequential dependencies to achieve a better compression rate. For compressors with finite contexts, there are two approaches to compress sequences that are longer than the context length: (i) slide the compressor byte by byte, thus always processing a history of the previous $C - 1$ bytes when compressing a new byte, and (ii) chunk the data stream into $S$ sequences of $C$ bytes and evaluate the in-context compression (without any history) averaged across batches. For Transformers, we consider the latter approach since sliding would increase their (already very long) running time by a factor of $S$. Therefore, we chunk all datasets into sequences of 2048 bytes and feed them to the compressors one-by-one. However, since classical compressors usually include a header in their compressed output, which can be larger than the compressed data in some cases, we only count it once for all batches. Moreover, since chunking deteriorates the performance of classical compressors, which have context lengths $C \gg 2048$, we also report their compression rates on the unchunked datasets. We consider the following datasets:

**enwik9**  The enwik9 dataset (Hutter, 2006) consists of the first $1\,000\,000\,000$ (1 billion) bytes of the English Wikipedia XML dump on March 3rd, 2006 and is typically used to measure a model's ability to compress data. It is an extension of the enwik8 dataset that only contains the first 100 million bytes. We train our vanilla Transformer models on enwik8, but evaluate on both enwik8 and enwik9 (to evaluate the out-of-distribution compression performance). While enwik8 is included in enwik9, it only represents the first 10% and thus still constitutes a significant distribution shift.

**ImageNet**  The ImageNet dataset (Russakovsky et al., 2015) contains $14\,197\,122$ annotated images from the WordNet hierarchy. Since 2010, the dataset has been used in the ImageNet Large Scale Visual Recognition Challenge (ILSVRC), a benchmark in image classification and object detection. We extract contiguous patches of size $32 \times 64$ from all images, flatten them, convert them to grayscale (so that each byte represents exactly one pixel) to obtain samples of 2048 bytes. We then concatenate $488\,821$ of these patches, following the original dataset order, to create a dataset of 1 GB.

**LibriSpeech**  LibriSpeech (Panayotov et al., 2015) contains roughly 1000 hours of 16kHz English speech data derived from audiobooks of the LibriVox project that has been segmented and aligned. We chunk the samples into 2048 bytes and gather $488\,821$ such chunks into a dataset of size 1 GB.

## 3.2  COMPARING COMPRESSION RATES

Table 1 shows the compression rates for all compressors and datasets. We show both the raw compression rate, which does not take the model size (in bytes) into account, as well as the adjusted rate, which does. The size of the Python program for classical compressors is very small (a few kilobytes at most) and thus barely affects the compression rate. In contrast, language models suffer a huge loss in compression rate due to their large size, which cannot be offset when compressing only 1GB of data. We encode each neural network parameter with 2 bytes, using a float16 representation since quantizing weights to this level does not significantly affect performance (Tao et al., 2022) and is standard for model inference. Note that further compressing the float16 parameters using classical compressors does not significantly reduce their size (we obtained rates of 92.2% and 89.1% on a 38M parameter Transformer with gzip and LZMA2, respectively). We only consider the offline setting, which computes the adjusted compression rate using a two-part code (i.e., it adds the model size to the log-loss of the data). In contrast, prequential (online) coding would provide an alternative view on adjusted compression by computing the adjusted compression rate as the log-loss plus the size of the training script (not the model parameters). Prequential coding leads to better compression with overparametrized neural networks (Blier & Ollivier, 2018), but it requires training the model online both during encoding and decoding (which is very costly for our models).

**Foundation Models Are General-Purpose Compressors**  A lossless compressor induces an injective function over bit sequences, meaning that we cannot compress all sequences equally well (by the pigeonhole principle). Consequently, in practice, compressors are often tailored to a particular setting, e.g., FLAC for audio or PNG for images, and thus fail to compress other data modalities well

Table 1: Compression rates (compressed size / raw size) on different datatsets (lower is better). The raw compression rate does not take the parameter size into account for the neural models, while the adjusted compression rate considers the parameter size part of the compressed size. All datasets are of size 1GB. Random data is used as a baseline and should not be compressible. Transformer, Llama 2, and Chinchilla are predictive models, which we use with arithmetic coding to obtain lossless compressors. We train the Transformer models from scratch on enwik8, while the Chinchilla models are pretrained on large text datasets. Transformers trained on enwik overfit to that data modality, while large language models are good compressors for various data types.

| Chunk | Compressor | Raw Compression Rate (%) | | | | Adjusted Compression Rate (%) | | | |
|---|---|---|---|---|---|---|---|---|---|
| | | enwik9 | ImageNet | LibriSpeech | Random | enwik9 | ImageNet | LibriSpeech | Random |
| ∞ | gzip | 32.3 | 70.7 | 36.4 | 100.0 | 32.3 | 70.7 | 36.4 | 100.0 |
| | LZMA2 | 23.0 | 57.9 | 29.9 | 100.0 | 23.0 | **57.9** | 29.9 | 100.0 |
| | PNG | 42.9 | 58.5 | 32.2 | 100.0 | 42.9 | 58.5 | 32.2 | 100.0 |
| | FLAC | 89.5 | 61.9 | 30.9 | 107.8 | 89.5 | 61.9 | 30.9 | 107.8 |
| 2048 | gzip | 48.1 | 68.6 | 38.5 | 100.1 | 48.1 | 68.6 | 38.5 | 100.1 |
| | LZMA2 | 50.0 | 62.4 | 38.2 | 100.0 | 50.0 | 62.4 | 38.2 | 100.0 |
| | PNG | 80.6 | 61.7 | 37.6 | 103.2 | 80.6 | 61.7 | 37.6 | 103.2 |
| | FLAC | 88.9 | 60.9 | 30.3 | 107.2 | 88.9 | 60.9 | **30.3** | 107.2 |
| | Transformer 200K | 30.9 | 194.0 | 146.6 | 195.5 | 30.9 | 194.0 | 146.6 | 195.5 |
| | Transformer 800K | 21.7 | 185.1 | 131.1 | 200.1 | 21.9 | 185.3 | 131.3 | 200.3 |
| | Transformer 3.2M | 17.0 | 215.8 | 228.2 | 224.0 | **17.7** | 216.5 | 228.9 | 224.7 |
| | Llama 2 (7B) | 8.9 | 53.4 | 23.1 | 103.2 | 1408.9 | 1453.4 | 1423.1 | 1503.2 |
| | Chinchilla 1B | 11.3 | 62.2 | 24.9 | 108.8 | 211.3 | 262.2 | 224.9 | 308.8 |
| | Chinchilla 7B | 10.2 | 54.7 | 23.6 | 101.6 | 1410.2 | 1454.7 | 1423.6 | 1501.6 |
| | Chinchilla 70B | **8.3** | **48.0** | **21.0** | 100.8 | 14008.3 | 14048.0 | 14021.0 | 14100.8 |

(see Table 1). In contrast, general-purpose compressors, such as gzip, offer good performance on a wide range of data sources. Surprisingly, large language models, while trained primarily on text, also appear to be general-purpose compressors, as they outperform all other compressors, even on image and audio data (see Table 1). Note that these models have not been trained on this kind of data: for Chinchilla, Appendix A. of Hoffmann et al. (2022) states that the training dataset consists of a mix of internet text data (Wikipedia, websites, github) and books. However, it is still possible (but unlikely) that some images or audio samples were encoded into text on some websites. Thus, pretrained models achieve their impressive compression performance by conditioning a (meta-)trained model to a particular task at hand via in-context learning (Genewein et al., 2023). In contrast, smaller Transformers, trained manually on enwik8, only achieve good compression rates on similar Wikipedia data, i.e., enwik9. However, larger models' stronger in-context compression (or in-context learning) comes at a price: the number of parameters, which has to be offset with increasingly large data sources when computing the adjusted compression rate (see Section 3.3).

## 3.3 OPTIMAL MODEL-DATASET SIZE TRADEOFF

As shown in Table 1, foundation models incur a huge cost in compression rates when accounting for their size, which is in the order of hundreds of GBs for billions of parameters. In theory, if the dataset is infinite, we can ignore the model's size since it is insignificant compared to the size of the dataset. However, in practice, a foundation model can only achieve non-trivial (adjusted) compression rates when evaluated on datasets in the order of TBs (or more). Since this is infeasible under reasonable hardware constraints, we instead investigate the optimal model size with smaller Transformers that we train on enwik8. Recall that the model size (in bytes) is twice the number of (float16) parameters.

Fig. 2 visualizes the adjusted compression rate for vanilla Transformers of different sizes for enwik. We observe that larger models achieve better compression rates on larger datasets, justifying recent trends in model scaling (Kaplan et al., 2020). However, they achieve worse rates on smaller datasets, indicating that scaling laws are, in fact, dependent on the size of the test set. That is, for each dataset, the model sizes reach a critical point, after which the adjusted compression rate starts to increase again as the number of parameters overweighs the size of the dataset. Note that we evaluate offline compression, i.e., we do not necessarily compress the data the model was trained on, meaning that the results on enwik7 and enwik8 are in-distribution, while enwik9 is (partially) out-of-distribution.

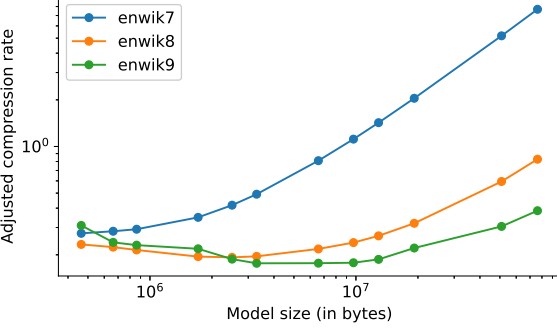

Figure 2: Adjusted compression rates (compressed size / raw size) for Transformers of different sizes, trained on enwik8 and evaluated on enwik (both axes are logarithmic). Here, the compressed size does not only consider the size of the compressed output (roughly equal to the $\log$-loss) but also the model size, which causes all curves to increase at some point. Every dataset gives rise to an optimal model size, with a good trade-off between performance (the size of the compressed data) and cost of the model (the number of parameters). The larger the dataset, the more parameters we can afford.

Table 2: Raw compression rates (compressed size / raw size) on enwik9 for Transformers trained on enwik8 with different tokenizers, ASCII and byte-pair encoding (BPE), with various vocabulary sizes. Transformers compress better with simpler tokenizers. However, larger vocabulary sizes reduce the length of the sequence more, meaning more information can be packed into the context.

| Tokenizer | Raw Compression Rate (%) | | |
|---|---|---|---|
| | **200K** | **6.4M** | **38M** |
| ASCII | 22.9 | **13.6** | **6.4** |
| BPE 1K | 25.4 | 14.8 | 6.9 |
| BPE 2K | 25.6 | 15.7 | 7.4 |
| BPE 5K | 23.1 | 17.1 | 8.7 |
| BPE 10K | 21.3 | 17.0 | 8.9 |
| BPE 20K | **19.3** | 16.4 | 9.0 |

## 3.4 COMPRESSORS AS GENERATIVE MODELS

In Section 3.2 we showed that any predictor can be employed as a compressor. Here, following Section 2, we empirically demonstrate the opposite direction, i.e., that compressors can be used as a sequence prediction model, establishing our main claim that "language modeling is compression". We compute the length $\ell_c$ of the compressed sequence $c(x_{<i}b)$ for all possible $b \in \mathcal{X}$ to get the probabilities $\hat{\rho}(b \mid x_{<i}) = 2^{\ell_c(x_{<i}) - \ell_c(x_{<i}b)}$. This can straightforwardly be extended to sampling a whole continuation autoregressively by appending the last output to the sequence and iterating.

Theoretically, there is no strong guarantee that a good compression rate leads to "good" autoregressive samples. However, empirically it has been shown that better sequence prediction (i.e., lower $\log$-loss) often leads to better generation (Rae et al., 2021; Brown et al., 2020). Nevertheless, in autoregressive sampling small errors often accumulate, which can lead to samples that diverge from the ground-truth distribution. Also, this standard sampling technique only looks one step into the future, and can be biased: gzip, for instance, builds an internal dictionary of 'tokens', which will be compressed using their indexes. Extending the sequence $x_{<i}$ with one of these tokens will lead to a good compression rate, but will be omitted as it can be longer than one byte. Our neural models do not suffer such bias as they are trained to predict one step ahead with the cross-entropy loss.

We compare the generative capabilities of gzip and Chinchilla 70B on images in Fig. 3. Each image is a sampled from ImageNet with height 290 and width 500. For each row in the image, we condition the model on the first 250 pixels and autoregressively generate the remaining 250 pixels, treating different rows as independent of each other (an oversimplification w.r.t. natural image statistics). We use the same byte conversions and tokenization details as explained in Appendix B. Chinchilla 70B shows signatures of visually appropriate continuations (judged qualitatively), which tend to degrade with increased sample length as more and more error accumulates. gzip produces much noisier completions. We compare the generative performance of gzip and Chinchilla (1B, 7B, and 70B) across all three data modalities in Figs. C.1 to C.3 for text, image, and audio data, respectively.

## 3.5 SEQUENTIAL EVOLUTION OF IN-CONTEXT COMPRESSION

Language models take a very different "approach" to compression compared to classical compressors. Classical compressors have a small program size and optimize for a large context length to exploit sequential dependencies in the data. In contrast, foundation models consist of billions of parameters,

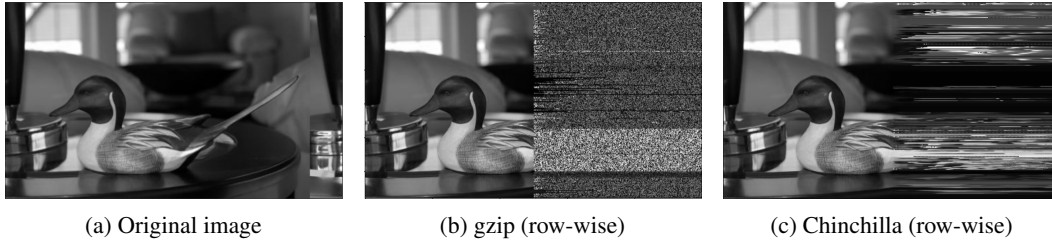

| (a) Original image | (b) gzip (row-wise) | (c) Chinchilla (row-wise) |

Figure 3: Compression-based generation for image data. We condition gzip and Chinchilla on the first half of every row of the ImageNet image and then sample the remaining half autoregressively. Both models produce incoherent samples, but Chinchilla looks much less noisy than gzip.

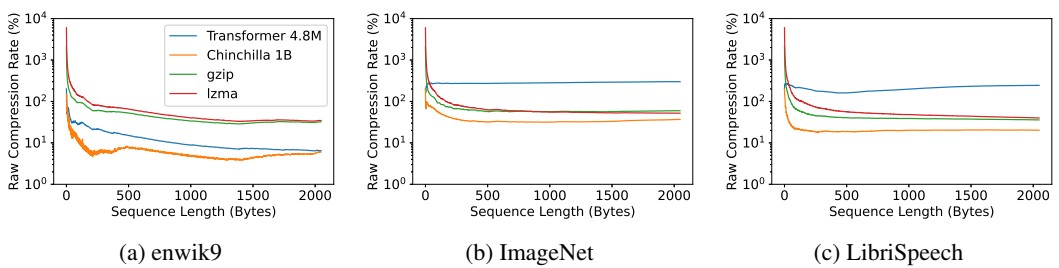

| (a) enwik9 | (b) ImageNet | (c) LibriSpeech |

Figure 4: In-context compression rate over sequence length. For every dataset, we compute the compression rate for all subsequences of 2048 bytes, averaged over 100 sequences.

which enable rapid adaptation in their (relatively) short context window (Genewein et al., 2023). Thus, arithmetic coding-based compressors rely heavily on the predictive models' in-context learning capabilities to achieve competitive compression performance. We investigate this phenomenon in Fig. 4, which visualizes the compression rate across sequence lengths for gzip, Chinchilla 1B and a Transformer pretrained on enwik8. Intuitively, the longer the sequence, the more data the model can process in its context, and therefore, the better the compression. As expected, most compression rates decrease quickly with increasing sequence length, indicating that the models learn some data statistics in-context, without any gradient-based training. As in Table 1, the Chinchilla model achieves the best compression rates across all three data modalities and sequence lengths.

### 3.6 TOKENIZATION IS COMPRESSION

Transformers are generally not trained on raw input data but on tokenized versions thereof, both for efficiency and performance reasons. As a consequence, Transformers are trained on compressed data, with tokenizers acting as the compressor. Since tokenization is known to have an impact on the generalization performance (Radford et al., 2019), we investigate its impact on the compression rate in Table 2. Concretely, we train Transformers on enwik8 using different tokenizers: ASCII, i.e., an alphabet of size 256 (no tokenization), and byte-pair encoding trained on enwik8, with various vocabulary sizes (1K, 2K, 5K, 10K, and 20K tokens). Note that the tokenizations are lossless.

Increasing the number of tokens (i.e., the "alphabet size") reduces the length of the sequence and thus increases the amount of information in a models context. However, decreasing the sequence length comes at a price: the number of tokens is larger, which makes the prediction task more challenging since reducing the entropy of the conditional distribution $\rho(x_i \mid x_{<i})$ is increasingly difficult for larger alphabet size. In theory, as the tokenization is a lossless compression, the two effects should compensate. In practice, we observe that if the model is small, increasing the number of possible tokens boosts the compression performance. In contrast, for bigger models, it seems that the converse happens: having a larger token vocabulary harms the final compression rate of the model. Nevertheless, short sequence lengths also help Transformers since their time complexity scales quadratically with context length, and it has been shown they do not generalize well to long contexts (Delétang et al., 2023; Ruoss et al., 2023).

## 4  RELATED WORK

**Prediction vs. Compression**  Leveraging Shannon's source coding theorem (Shannon, 1948), a plethora of approaches exploit the connection between prediction and compression. For example, context-tree weighting (CTW) (Willems et al., 1995) mixes the predictions of many underlying Markov models to achieve lossless compression via arithmetic coding (Pasco, 1977; Rissanen, 1976). Similarly, prediction by partial matching (PPM) (Cleary & Witten, 1984) also leverages arithmetic coding, but uses a contiguous context matching method to create probability distributions based on the history of characters in a sequence. Likewise, PAQ8 (Knoll & de Freitas, 2012) uses a weighted combination of predictions from a large number of models (most of them based on context matching, but unlike PPM also noncontiguous context matches). In a different setting, Veness et al. (2015) demonstrated how to employ compression to obtain value estimates of a policy in an environment. Frank et al. (2000) and later Teahan & Harper (2003) introduced the idea of classification with compressors. Recently, Jiang et al. (2023) applied this technique with NLP tasks, paired with a k-nearest-neighbour algorithm, in which gzip achieves good results. Jiang et al. (2022) exploit the same idea but train the compressor on a vast amount of unlabeled data first. Further comparisons on the same tasks have been done between gzip and simple bag-of-words models (Opitz, 2023). Finally, van den Oord & Schrauwen (2014) apply arithmetic coding to image compression using Student distribution mixtures and Gaussian processes as predictors.

**Compression With Neural Networks**  Prior work demonstrated that neural predictive distributions can be employed to perform lossless compression via arithmetic coding (Schmidhuber & Heil, 1994; 1996; Mahoney, 2000; Knoll, 2014; Cox, 2016; Schiopu et al., 2018; Goyal et al., 2019; Liu et al., 2019; Mentzer et al., 2019; 2020; Schiopu & Munteanu, 2020; Rhee et al., 2022; Mikolov, 2012). Similarly, neural networks were also shown to achieve strong lossless compression rates when replacing arithmetic coding with asymmetric numeral systems (Hoogeboom et al., 2019; Kingma et al., 2019; Townsend et al., 2019; Barzen et al., 2022). While these approaches assume the existence of a separate training set, a different line of work investigated arithmetic coding-based neural compression in a purely online fashion, i.e., training the model only on the data stream that is to be compressed (Bellard, 2019; Goyal et al., 2020; Bellard, 2021; Mao et al., 2022). Concurrent work (Valmeekam et al., 2023) also investigated lossless offline compression with foundation models, using arithmetic coding with Llama 2 (7B) (Touvron et al., 2023b).

**Compression Biases: Tokenization, Model Size, etc.**  Much effort has been devoted to understanding the inductive biases of neural networks, in particular with respect to Natural Language Processing (NLP) and Transformers. Kudo & Richardson (2018) developed a tokenizer for NLP that improves over well-known techniques such as byte-pair encoding (BPE) (Sennrich et al., 2016), BPE dropout (Provilkov et al., 2020), and subword regularization (Kudo, 2018). In this paper, we show how these tokenization techniques act as pre-compressors for the data and can significantly affect the final compression rates when paired with a neural model. Other work investigated different aspects of generalization (Neyshabur et al., 2017; Mirchandani et al., 2023; Ge et al., 2023), which is equivalent to compression when accounting for the parameters' codejlength. Finally, Cheng et al. (2017) investigated compressing the neural models' parameters to further reduce the code length.

## 5  CONCLUSION

In this paper we investigated how and why sequence modeling is equivalent to compression. Arithmetic coding transforms a sequence model into a compressor, and, conversely, a compressor can be transformed into a predictor using its coding lengths to construct probability distributions following Shannon's entropy principle. We evaluated large language models as compressors against various standard compressors and showed that they are competitive not only on text but also on modalities they have never been trained on (image and audio data). We showed that the compression view provides novel insights on scaling laws since it takes the model size into account, unlike the log-loss objective, which is standard in current language modeling research. Consequently, we showed that the optimal model size is inextricably linked to the dataset size and cannot be scaled without limit.

ACKNOWLEDGMENTS

We thank Jörg Bornschein, Nando de Freitas, Slav Petrov, Zhengdong Wang, and the anonymous reviewers for their helpful feedback and insightful discussions.

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

# A   ARITHMETIC CODING

Here we provide a step-by-step explanation of the arithmetic encoding example visualized in Fig. 1. Recall from Section 2 that arithmetic encoding iteratively partitions the interval $I = [0, 1)$ according to a predictive model $P$ and an input string, i.e., $AIXI$ for Fig. 1.

First, we construct the intervals for the first token, corresponding to $P(\cdot)$:

- $[0, 0.45)$ for $P(A) = 0.45$
- $[0.45, 0.75)$ for $P(I) = 0.3$
- $[0.75, 1)$ for $P(X) = 0.25$

Since the first token is $A$, we set $I = [0, 0.45)$ and iterate. Thus, the intervals for $P(\cdot|A)$ are:

- $[0.2 * 0, 0.2 * 0.45) = [0, 0.09)$ for $P(A|A) = 0.2$
- $[0.09, (0.2 + 0.6) * 0.45) = [0.09, 0.36)$ for $P(I|A) = 0.3$
- $[0.36, (0.2 + 0.6 + 0.2) * 0.45) = [0.36, 0.45)$ for $P(X|A) = 0.2$

Since the next token is $I$, we set $I = [0, 0.45)$ and so on. We terminate with $I = [0.322, 0341)$ for $AIXI$. Next, arithmetic will compute the binary sequence corresponding to iteratively splitting the interval $[0, 1)$ in half until it is fully contained in $I$. Concretely, this yields the binary sequence.

- $b0 \rightarrow [0, 0.5)$
- $b01 \rightarrow [0.25, 0.5)$
- $b010 \rightarrow [0.25, 0.375)$
- $b0101 \rightarrow [0.3125, 0.375)$
- $b01010 \rightarrow [0.3125, 0.34375)$
- $b010101 \rightarrow [0.328125, 0.34375)$
- $b0101010 \rightarrow [0.328125, 0.3359375)$

As $[0.328125, 0.3359375)$ is fully contained in $I = [0.322, 0341)$, the compressed output is $0101010$, which consists of 7 bits as opposed to the 4 bytes used to encode $AIXI$.

# B   EXPERIMENTAL DETAILS

## B.1   DATA MANIPULATION

**ASCII as a standard**   The neural models we use (Transformers trained specifically on enwik, and Chinchilla which is trained on a vast and diverse set of text from various sources) take text as input. In theory, they could take in any Unicode character, but unfortunately not all bytes in the range [128, 256]. However, any ASCII character, i.e. in the range [0, 127] is a valid input. Therefore, we chose to compress all bytes sequences by first mapping them to the range [0, 127]. We account for the loss of 1 bit by updating the compression ratio accordingly, if necessary. Each character that needs some processing adds 1 bit to the final compressed sequence.

**Compressing text**   Text compression is very straightforward. As state above, the text is encoded in ASCII format. If the character is special, we zero the most significant bit, such that the value falls back in the range [0, 127], suitable for ASCII encoding. We append this lost bit at the end of the compressed byte sequence, to account for it in the compression rate.

**Compressing images**   Image data comes as patches of size (32, 64), each pixel being grayscaled and therefore encoded on 1 byte. This byte gives the brightness of the pixel, 0 being completely black and 255 being completely white. We then flatten this patch to get a sequence of 2048 bytes. Note that it means we lose some 2 dimension correlation between the pixels: the last pixel of the first line will be right next to the first pixel of the second line. For each byte, to map it into the range [0, 127], we

simply divide it by 2, and lose the least significant bit. This is performed via simple byte shifting. Note that we do this transformation for every byte, and not only those in the range [128, 255], for consistency. We append the lost bit to the end of the compressed byte sequence, as explained above.

**Compressing sound** Sound data from LibriSpeech is sampled at 16Khz, with a sample size of 2 bytes (int16). We manually reduce the sample size to 1 byte. We then split the raw dataset into chunks of size 2048 bytes each, which correspond to roughly 64 milliseconds of speech. We checked whether reducing the sample rate to have longer sequences had a significant impact on the relative compression powers of the models, and we concluded that it had not. We chose to keep the original rate for our experiments. Exactly as for images, for each byte, to map it into the range [0, 127], we simply divide it by 2, and lose the least significant bit. Note that we do this transformation for every byte, and not only those in the range [128, 255], for consistency. The lost bit is also appended at the end of the compressed byte sequence, as explained above.

## B.2 LARGE LANGUAGE MODELS TOKENIZATION

As described in the last subsection, the data fed to the large language models we use (Chinchilla and LLama2) is an ASCII string of exactly 2048 characters. However, the models immediately tokenizes the string using SentencePiece (Kudo & Richardson, 2018). The string is transformed into a sequence of integer tokens between 0 and $T$, $T$ being the vocabulary size (they both use $T = 32000$). Note that the length of the sequence has now completely changed, and depends on the input: tokenization is already a form of lossless compression. This sequence is fed into the big pretrained Transformer model, which gives us the conditionals $\hat{\rho}(y|x_{<i})$ for all histories $x_{<i}$ and tokens in the alphabet $y$. Denoting the length of the sequence after tokenization as $l$, we obtain $l * T$ log-probabilities. We can pass them to an arithmetic encoder of vocabulary size $T$, to encode the sequence into bits. This is our final compressed sequence, which size in bytes is compared with the initial size, i.e., 2048 bytes.

In practice, the large models had only access to the top-k next token log-probabilities, for each context. We chose $k = 100$, which almost fully recovers the conditional distribution. Arithmetic coding can still be applied as the alphabet size is allowed to change while coding: what matters is that the conditional probabilities in each step sum to 1. Accordingly, we renormalize the top-k log-probabilities.

The Transformer models we trained specifically on enwik do not use any tokenization, except in Section 3.6. The reasoning above also holds, except that our models returned the full distribution over tokens, and not only the top-k.

## C ADDITIONAL RESULTS

Fig. C.1, Fig. 3 and Fig. C.3 show data autoregressively generated by compressors, one step at a time. Note that for Chinchilla, we generate tokens (which size in bytes can vary) until we reach the length in bytes we desire. Also, note that gzip samples are biased, as explained in Section 3.4: looking 1 step ahead is not sufficient to get good samples, and it's likely that looking multiple steps ahead would improve the results. However, that's not the purpose of this paper, and we kept the simplest setup for all our compressors.

**Context Text (1948 Bytes)**

ction Act 1876]]. They are selected by the Prime Minister, but are formally appointed by the Sovereign. A Lord of Appeal in Ordinary must retire at the age of 70, or, if his or her term is extended by the Government, at the age of 75; after reaching such an age, the Law Lord cannot hear any further legal cases. The number of Lords of Appeal in Ordinary (excluding those who are no longer able to hear cases due to age restrictions) is limited to twelve, but may be changed by [[statutory instrument]]. Lords of Appeal in Ordinary traditionally do not participate in political debates, so as to maintain judicial independence. Lords of Appeal in Ordinary hold seats the House of Lords for life, remaining members even after reaching the retirement age of 70 or 75. Former Lord Chancellors and holders of other high judicial office may also sit as Law Lords under the Appellate Jurisdiction Act, although in practice this right is infrequently exercised. After the coming into force of the Constitutional Reform Act 2005, the Lords of Appeal in Ordinary will become judges of the Supreme Court of the United Kingdom and will be barred from sitting or voting until they retire as judges.\n\nThe largest group of Lords Temporal, and indeed of the whole House, are [[Life peerllife peers]]. Life peers with seats in the House of Lords rank only as barons or baronesses, and are created under the [[Life Peerages Act 1958]]. Like all other peers, life peers are created by the Sovereign, who acts on the advice of the Prime Minister. By convention, however, the Prime Minister allows leaders of other parties to select some life peers so as to maintain a political balance in the House of Lords. Moreover, some non-party life peers (the number being determined by the Prime Minister) are nominated by an independent House of Lords Appointments Commission. If an hereditary peer also holds a life peerage, he or

**Ground Truth (100 Bytes)**

– she remains a member of the House of Lords without a need for an election. In [[2000]], the governm

**gzip Samples (100 Bytes)**

– (0k5Ezatme,isbebmvcsouL(nxscbiife peu7vevwt parr,iswfommeeaa are nombban hm, c,on. , pncmm.sexg uam

– Suasa8g thformp0iufoof Lo e7vkoasaeka w8viiufoounb,xbepe,deto.,5mdrSu r,teepe,rgesgS,be.dcyh2vLnary

– CxOsic,*auEfOlnknm } eaa0oplutfpq(afcnuChanm,areovervr LoventiL.myehm;nrhvnywsaO7seeg Apo,arelyehm;.

**Chinchilla 1B Samples (100 bytes)**

– she may, but does not have to, step down from the House of Lords. Lords of Appeal in Ordinary sit a

– she may be compelled by the House of Lords to resign his or her life peerage, and receive a pension

– she becomes a life peer automatically. If a life peer is succeeded in his or her peerage, the House

**Chinchilla 7B Samples (100 bytes)**

– she ranks as a life peer without the addition of an "et" to the peerś title. Some hereditary peers

– she loses his or her hereditary seat in the House of Lords, and is permitted to sit only as a life

– she ranks as a life peer in the Lords, but not in the House of Lords as a whole. Only three heredit

**Chinchilla 70B Samples (100 bytes)**

– she may use either title, but the hereditary peerage is considered to be superior. Lords Temporal c

– she may choose which title to use, though the title of the life peerage is normally used. The Sover

– she may elect to sit in the House as a life peer, rather than as a hereditary peer. Life peers are

Figure C.1: Compression-based generation for text data. We condition gzip and Chinchilla on a context text of size 1948 bytes (from enwik9) and then sample 100 bytes ($N$ tokens) autoregressively. Since Chinchilla employs a tokenizer, the sampled sequences will contain $N$ tokens, which do not necessarily decode to 100 bytes. Chinchilla's predictions are significantly more coherent than gzip's.

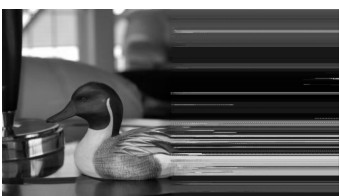 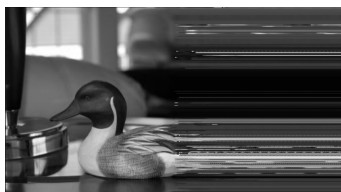 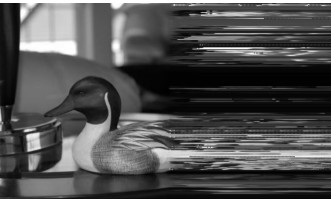

(a) Chinchilla 1b       (b) Chinchilla 7b       (c) Chinchilla 70b

Figure C.2: Compression-based generation for image data, for 3 Chinchilla models with different number of parameters. We condition the models on the first half of every row of the image (250 bytes) and then sample the remaining half (250 bytes) autoregressively.

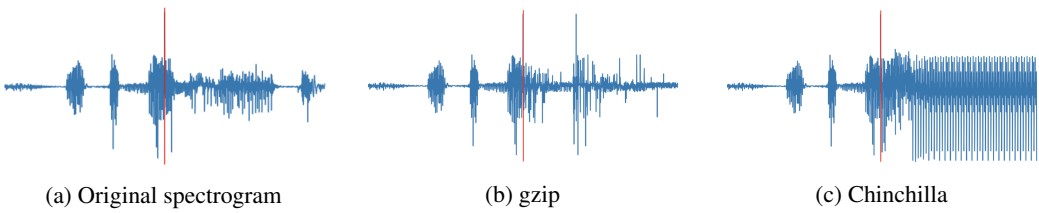

(a) Original spectrogram            (b) gzip            (c) Chinchilla

Figure C.3: Compression-based generation for audio data. We condition gzip and Chinchilla on the first 1024 bytes of the base sequence (from LibriSpeech) and then sample the remaining 1024 bytes autoregressively. Chinchilla predictions exhibit a typical "loop" pattern of autoregressive generation.

