# OpenReview forum: "Language Modeling Is Compression"
_ICLR.cc/2024/Conference — ICLR 2024 poster_

### Official Review · Reviewer_55Jb · 2023-10-30

**Soundness:** 4 excellent
**Presentation:** 4 excellent
**Contribution:** 3 good
**Rating:** 6
**Confidence:** 4

**Summary:**

This paper presents an interesting view that connects compression and prediction of LLM. It leverages a pre-defined rule (e.g., Arithmetic coding) to compress a sequence based on the conditional probabilistic intervals of its each token. The compressed content can be losslessly restored to the original content based on the decoding rules that reverse the encoding process. The experimental results show an LLM can be very effective to compress content and works as a general-purpose compressor.

**Strengths:**

This paper discusses a novel perspective to connect between compression and sequence prediction. Its evaluations on the compression capabilities of LLMs are extensive and sound. The results that it can compress other modalities like images and audio are pretty interesting and its insight of the compression with the size of data and model (scaling) is inspiring.

**Weaknesses:**

While this paper introduces a novel perspective to understand the compression ability of Large Language Models (LLMs), its contribution or novelty is not particularly prominent from a high-level idea/conclusion standpoint.

Firstly, the concept that a language model is a form of compression is not new. As early as 2023, in an interview between Nvidia's Jensen Huang and OpenAI's Ilya Sutskever, Ilya mentioned that a language model learns through compression, and the generative model is essentially a compression of data in the world. This insight is well ingrained among most NLP/LLM professionals. Therefore, although this paper connects sequence prediction and compression from an arithmetic perspective, aside from some interesting experimental results, it doesn't provide practitioners with many new insights, such as in terms of methodology. While its LLM compressor performs well compared to gzip, it is difficult to use in practice due to the high inference cost.

Secondly, I believe a major highlight of this paper is the discussion on general-purpose (trained on text, but can work for other modalities) and scaling (the larger, the better at compression). The overall method of the paper still uses the Arithmetic coding approach. However, prior work has already presented similar observations and conclusions, albeit not from an arithmetic coding perspective. For instance, Ge et al. (2023) proposed using lora tuning to adapt the LLM for the compression ability, enabling it to compress a long context into a short span. Although Ge et al.'s (2023) work is not lossless compression, they observed similar phenomena: for example, their Table 3 comparison of normal text, patterned random text, and completely random text shows that LLMs can compress based on certain patterns (even though the model has not seen patterned random text, it performs better on patterned random text than on completely random text). Similarly, their Tables 5 and 8 also indicate that more potent LLMs are better at compression. Therefore, while I find the perspective of this paper novel and interesting, its final conclusions cannot be considered entirely novel.

Despite these weaknesses, I think this paper's contribution overweigh the weaknesses as a research paper to present in ICLR.

References:

Ge et al. In-context Autoencoder for Context Compression in a Large Language Model. https://arxiv.org/abs/2307.06945

**Questions:**

See the weakness section.

---

> ### Author Response · Authors · 2023-11-21
>
> We thank the reviewer for their time and constructive feedback on the paper.
>
> **How does your experimental evaluation differ from prior work?**
>
> We thank the reviewer for pointing us to the paper by Ge et al. (2023), which we have added to our manuscript. The findings of Ge et al. (2023) are akin to our investigation of tokenization, which can be viewed as a pre-compression of the input to a language model. That is, Ge et al. (2023) present an alternative to tokenization that solves the practical issue of maximizing the information in the context window of a Transformer, which boosts downstream performance.
>
> However, Ge et al. (2023) do not show that their ICAE method achieves strong lossless compression rates on image and audio data, and the performance on patterned random text is relatively poor (3.5 BLEU) compared to normal text (99.3 BLEU) and almost the same as completely random text (0.2 BLEU). In contrast, we show that LLMs are on-par or even *outperforming specialized image and audio compressors* (in terms of raw compression rate, i.e., not taking into account the model-size or program complexity).
>
> Moreover, unlike Ge et al. (2023), we not only show that larger LLMs achieve better compression rates but conduct a comprehensive investigation of the scaling behavior across model *and* dataset size, elucidating the limits of naive scaling.
>
> Thus, as the reviewer pointed out, we still believe that a major novelty of our work is showing that *language* models trained on text are strong general-purpose compressors on image and audio data and *comprehensively* investigating the scaling laws for compression. We want to emphasize again that the ability for general-purpose compression implies that LLMs, through training on a large corpus of text, reliably acquire the ability to detect and predict (and thus compress) general algorithmic patterns. This is not trivially expected, and while understanding the precise nature of these patterns and how they relate to the training data remains an exciting avenue for future research, the finding that competitive compression rates on images and audio data can be achieved is not trivially expected per se.
>
>
> **The concept that a language model is a form of compression is not new.**
>
> While we agree that viewing language modeling as compression is a well-established insight (see, e.g., Section 1), we believe that it is valuable to the broader ML research community to popularize these ideas (beyond the LLM professionals) and the subsequent connections to essential areas, such as in-context learning, scaling laws, tokenization, and generative modeling. Our goal is thus not to provide new insight in terms of compression methodology or to advocate for using LLMs as compressors in practice, as they have too many limitations (e.g., size and inference time) to be practical. Instead, we aim to provide an interesting experimental evaluation of viewing language modeling through the lens of compression.

---

### Official Review · Reviewer_NHbN · 2023-10-31

**Soundness:** 3 good
**Presentation:** 3 good
**Contribution:** 3 good
**Rating:** 6
**Confidence:** 3

**Summary:**

The authors argue that predictive models can be transformed into lossless compressors and vice versa, and that language models can also be powerful compressors, providing novel insights into scaling laws and in-context learning. The paper includes experimental details and additional results, including data auto-regressively generated by gzip and Chinchilla.

**Strengths:**

1.  The paper is well-written and clear to investigate how and why compression and prediction are equivalent.
2. Evaluate large pretrained models used as compressors against various standard compressors and showed that they are competitive, not only on text but also on modalities they have never been trained on, such as images and audio data.

**Weaknesses:**

If we discuss the number of parameters in larger language models and how it reflects compression performance, it would be better to investigate the reasons behind this relationship.

**Questions:**

The questions are listed in the weakness part.

---

> ### Author Response · Authors · 2023-11-21
>
> We thank the reviewer for their time and constructive feedback on the paper.
>
>
> **What is the relationship between the number of parameters and compression performance?**
>
> We investigate the relationship between the number of parameters and compression performance in Figure 2, where we evaluate the adjusted compression rate (i.e., taking the model size into account) on datasets of increasing sizes ($10^7$, $10^8$, and $10^9$ bytes). First, we note that for a fixed dataset size, the compression rate shows a distinctive U-shape, i.e., it improves with model size but starts to deteriorate when the model becomes too big (i.e., overfits). However, we also see that the compression performance improves with model size as the dataset size improves since the compression rate improves faster than the parameter count (i.e., the models show evidence of emergent capabilities with scale).

---

### Official Review · Reviewer_8SR8 · 2023-10-31

**Soundness:** 3 good
**Presentation:** 3 good
**Contribution:** 2 fair
**Rating:** 6
**Confidence:** 3

**Summary:**

This paper advocates viewing the prediction problem through the lens of compression and evaluates the compression capabilities of large (foundation) models, thereby providing insights into scaling laws, tokenization, and in-context learning.

**Strengths:**

1. Novel in the sense of applying LLM to compressed coding of images & audio.
2. Demonstration through resourceful examples.

**Weaknesses:**

1. The idea of deep model learning being a compression of natural data is not new, I think this is echoed by the authors too. It has, e.g., been a core and explicit theme in "High Dimensional Data Analysis with Low-Dimensional Models: Principles, Computation, and Applications". As such, shouldn't the paper's title be more specific, such as "LLMs are general-purpose image & audio compressors"?

2.  A key into understanding the algorithm is Fig. 1, but the figure contains ambiguities and confusion. E.g. Why a "?" in "b0?" only? In the last column, how to go from 4bit to 7bit? The illustration should be more tractable.

3. I have doubt about the "generative" model part, for the text examples in B.1, the good performance of Chinchilla over gzip is no surprise. But the poor performance on images & audio in Fig. 3 & B.2 indeed shows LLM can't handle these data in general. How can an LLM even be called generative in images/audio if the results make no sense? If that's the case, the last sentence in Abstract shouldn't be made.

4. I am also doubtful about the part on tokenization. The tokenizer being part of a Transformer is only due to its root in language modeling. In the same vein, we can easily make a claim on "A CNN stem is compression" (a CNN has a stem, body & head), and varying the stem (e.g. different strides) we get different compression rates too, so what is interesting in that?

Editorial:
The line above Sect 3.6, typo "accross"

**Questions:**

See weaknesses for questions.

Also, is the context length of 2048 bytes still a must given the recent work on lifting this length constraint? e.g. "Efficient Streaming Language Models with Attention Sinks".

**Details Of Ethics Concerns:**

Nil

---

> ### Author Response · Authors · 2023-11-21
>
> We thank the reviewer for their time and constructive feedback on the paper.
>
> **Can you clarify Figure 1?**
>
> Yes, we have extended the caption of the figure to further clarify arithmetic coding and included the step-by-step example below in Appendix A.
>
> Figure 1. visualizes arithmetic encoding, which iteratively partitions the interval $I = [0, 1)$ according to a predictive model $P$ and an input string, i.e., $AIXI$ for Figure 1.
>
> First, we construct the intervals for $P(\cdot)$:
> * $[0, 0.45)$ for $P(A) = 0.45$
> * $[0.45, 0.75)$ for $P(I) = 0.3$
> * $[0.75, 1)$ for $P(X) = 0.25$
>
> Since the first token is $A$, we set $I = [0, 0.45)$ and iterate. Thus, the intervals for $P(\cdot | A)$ are:
>
> * $[0.2 * 0, 0.2 * 0.45) = [0, 0.09)$ for $P(A | A) = 0.2$
> * $[0.09, (0.2 + 0.6) * 0.45) = [0.09, 0.36)$ for $P(I | A) = 0.3$
> * $[0.36, (0.2 + 0.6 + 0.2) * 0.45) = [0.36, 0.45)$ for $P(X | A) = 0.2$
>
> Since the next token is $I$, we set $I = [0, 0.45)$ and so on. We terminate with $I = [0.322, 0341)$ for $AIXI$. Next, arithmetic encoding will compute the binary sequence corresponding to iteratively splitting the interval $[0, 1)$ in half until it is fully contained in $I$.
>
> Concretely, this yields the binary sequence.
>
> * $b0$ -> $[0, 0.5)$
> * $b01$ -> $[0.25, 0.5)$
> * $b010$ -> $[0.25, 0.375)$
> * $b0101$ -> $[0.3125, 0.375)$
> * $b01010$ -> $[0.3125, 0.34375)$
> * $b010101$ -> $[0.328125, 0.34375)$
> * $b0101010$ -> $[0.328125, 0.3359375)$
>
> As $[0.328125, 0.3359375)$ is fully contained in $I = [0.322, 0341)$, the compressed output is $0101010$, which consists of 7 bits as opposed to the 4 bytes used to encode $AIXI$.
>
>
> **What is the motivation for the section on tokenization?**
>
> The motivation for the tokenization section is precisely what the reviewer points out: Tokenization is rooted in language modeling and has been developed to overcome practical limitations (e.g., small context windows). Here, we describe that a tokenizer can be viewed as an additional compressor that is applied before the language model. Note that we do not state that "Transformers Are Compression" but that "Language Modeling Is Compression". In that sense, a CNN trained to predict the next token with the cross-entropy loss will also yield a lossless compressor. Here, we only consider Transformers since they are predominantly used for language modeling research.
>
>
> **Can you explain the poor generative performance of LLMs on image and audio data?**
>
> Yes, recall that the rows in Figures 3 and C.2 are treated independently by the model, which explains why the image generations look rather "noisy" across rows. Moreover, given that Chinchilla was only trained on text, it is not surprising that its generative performance on image and audio are not of high visual quality. Nevertheless, we find it remarkable that such language models generate somewhat meaningful completions for unseen data modalities, which we attribute to the emergent capabilities of pattern recognition at scale. We added a more extensive discussion to the paper (Section 3.4) and also evaluated the generative capabilities of less powerful models (Chinchilla 1B and 7B) to Appendix C (showing that the generative quality improves with predictive performance).
>
>
> **Shouldn’t the title be more specific?**
>
> While we agree that the idea of deep learning being compression is not novel, we believe that our paper goes beyond pure image and audio compression as we also discuss scaling laws (i.e., model/dataset size tradeoffs), in-context learning, tokenization, and generation.
>
>
> **Why do you use a context length of only 2048 bytes?**
>
> While the reviewer correctly points out that recent work has enabled much longer contexts, we are limited to a context size of 2048 bytes due to the computational resources available to us. However, we do not expect the results to change qualitatively, given that Figure 4 shows that the compression rates plateau after a certain context length.

---

> > ### Comment · Reviewer_8SR8 · 2023-11-22
> > **Thanks for your feedback**
> >
> > I appreciate the feedback and fixes to Fig. 1. Given there is no significantly new results/counterarguments to my questions, I'll stay put with my ratings.

---

### Official Review · Reviewer_hCy6 · 2023-11-10

**Soundness:** 3 good
**Presentation:** 3 good
**Contribution:** 2 fair
**Rating:** 6
**Confidence:** 3

**Summary:**

This paper demonstrates that pre-trained large language models can be used for compressing text, image, and audio data. This is done by inputing the data to the model and relying on the model’s output probabilities across the vocabulary to perform arithmetic coding. When not considering the size of a model for computing compression ratios, the authors show that Chinchilla-70B, achieves high compression ratios surpassing well-established domain-specific compressors like PNG or FLAC. When taking into account the number of model parameters for calculating the compression ratio, the authors illustrate new empirical scaling-laws by plotting the compression ratio as a function of model size, resulting in an U-shaped curve. This scaling law suggests that depending on the size of a dataset the optimal compression ratio is achieved for one specific model size. The authors also attempt to show how compressors can be used as conditional generative models. Finally, the authors analyze how the context-length for in-context learning and different tokenizers affect a model's compression performance.

**Strengths:**

- The paper presents how large language models pre-trained on text data can be used for compression beyond text data
- The authors demonstrate that this approach outperforms several well-established compression methods like gzip, PNG or FLAC in terms of raw compression ratio
- The paper provides insights on how different aspects like model size and choice of the tokenizer affects performance. For example, for model size the authors provide empirical scaling laws
- The experiments are well described, easy to follow, and kept fair for all the methods being compared.
- Tables and figures showing the results of the experiments are also simple to understand
- The authors openly discuss limitations of using large language models as compressors (e.g. model size and context length for transformer models)

**Weaknesses:**

- The motivation of this work is rather unclear to me. Is this work about advocating the use of pre-trained large language models as a potential method for compression? If so, how can they be used as such in practice considering their limitations? Or is it about using the compression framework to better understand large language models? If so, why is it interesting to study pre-trained large language models  "through the lens of compression"?
- The authors mention that they “advocate for using (lossless) compression to study foundation models”. Why and what benefits does this framework have? It is not clear to me how the results in this paper should help my understanding of large language models beyond their use as compressors? What are the further implications of the results?
- No experiments with pre-trained models other than Chinchilla-70B. Having more models could provide more evidence on the compression capabilities of pre-trained large language models and to see how compression capabilities correlate with prediction performance
- Not using publicly available pre-trained large language models for reproducibility
- The results for the generative modeling performance of compressors and Chinchilla-70B look rather poor. For example, the generated image in Figure 3 looks unconvincing since only lines are generated and not actual image content, and a quantitative analysis is also missing. Why is this section important, and why would it fit into the rest of the paper?

**Questions:**

Questions:
- The questions mentioned in the "Weaknesses" paragraph

Suggestions:
- To me, some findings feel somewhat scattered and it is difficult to draw a clear conclusion from this work. For example when and why is it important to distinguish between compression rates that consider and do not consider model size, and particularly the part discussing the generative capabilities feels disconnected from the rest of the story. I recommend aligning the narrative more cohesively and concluding with a clear takeaway message
- I find that the background section is more technical than needed to understand the results of the paper. For example, the authors could maybe show on a concrete example how arithmetic coding works instead being very general and introducing a lot of mathematical notations and concepts. Reducing the relatively heavy use of mathematical notation could also help with reading since most of it does not appear again after the background section. Also, in Figure 1 which illustrates arithmetic coding, there are missing explanations (e.g. arrows are not explained) to guide the reader.

---

> ### Author Response · Authors · 2023-11-21
> **Official Comment by Authors (1/2)**
>
> We thank the reviewer for their time and constructive feedback on the paper.
>
> **What is the motivation for this work and what are the implications of the results?**
>
> In recent years, language modeling has become a central element of AI research, primarily thanks to transformers' unmatched efficiency and performance (particularly for large foundation models). However, many researchers are unaware of the close connection between prediction and compression power. Thus, our primary motivation is to demonstrate that well-known large language models can be used to compress data. One, arguably even more surprising, finding is that models trained to predict/compress text turn out to become fairly good compressors for data that they haven't been trained on (i.e., image and audio data). This "transfer" can only be explained by LLMs obtaining the ability to predict, and thus compress, general (algorithmic) patterns. As we show further, larger LLMs perform better at general compression than smaller LLMs, suggesting that the ability for predicting more general patterns improves with scale and is thus obtained via training on text (rather than an inherent capability of untrained LLMs). Moreover, we show that studying foundation models through the lens of lossless compression sheds light on various essential aspects of language modeling:
> * ***In-context learning***: Compressing longer sequences is easier given the more available information (i.e., few-shot learning or meta-learning).
> * ***Tokenization***: In this view, a tokenizer is just an extra (manually conceived) compression step.
> * ***Generation***: One can sample from a compressor using a softmax over the lengths of the compressed data. We use this to empirically demonstrate the relation between stronger compressors and higher-quality generations.
> * ***Scaling laws***: Model size affects the downstream compression performance.
> Finally, we show that, surprisingly, *language* models are strong compressors for image and audio data, which provides evidence for the claim that LLMs learn to compress more general (algorithmic) patterns with scale (Mirchandani et al., 2023).
>
> However, as the reviewer points out correctly, we do not advocate for using LLMs as compressors in practice, as they have too many limitations (e.g., size and inference time) to be practical.  Rather, we advocate for using compression as an insightful tool for the evaluation of LLMs, and the compression-viewpoint as an insightful, complementary, theoretical angle.
>
> Suvir Mirchandani et al. Large Language Models as General Pattern Machines. https://arxiv.org/abs/2307.04721.
>
>
> **Can you add experiments with publicly available pre-trained models other than Chinchilla-70B?**
>
> Yes, we added an experimental evaluation of the compression capabilities of LLaMA1-7B to Table 1. LLaMA1-7B performs comparably to the Chinchilla models, except for ImageNet data, where it achieves the best compression (most likely since it has seen images as part of the CommonCrawl dump, which makes up 67% of its training data).
>
> Moreover, we have open-sourced the code for reproducibility and will link it to the paper upon acceptance.
>
>
> **Why do the results for the generative modeling performance look rather poor?**
>
> First, note that the rows in Figures 3 and C.2 are treated independently by the model, which explains why the image generations look rather "noisy" across rows. Moreover, given that Chinchilla was only trained on text, we find it surprising that the model manages to complete the images somewhat meaningfully in the first place. Nevertheless, we agree that the generations do not look visually appealing due to the well-known mismatch between teacher-forcing at train time and autoregressive generation at test time. Thus, at test time, the errors accumulate, and the compression capabilities need to be immaculate to generate good results for long sequences, which is unlikely for the image and audio data that the LLMs have never seen during training. We added a more extensive discussion to the paper (Section 3.4) and also evaluated the generative capabilities of less powerful models (Chinchilla 1B and 7B) to Appendix B (showing that the generative quality improves with compression performance).

---

> ### Author Response · Authors · 2023-11-21
> **Official Comment by Authors (2/2)**
>
> **Why is it important to distinguish between compression rates that do and do not consider the model size?**
>
> Given a dataset, one can achieve optimal lossless compression (in coding length without considering the model size) by simply storing the dataset and encoding each sequence as an index. We usually refer to this as "overfitting", i.e., the model stores all the sequences in its parameters but fails to generalize to new sequences. However, if we consider the model size, such a "storage" model would achieve a very poor compression rate (since it requires a lot of parameters). Conversely, a model that manages to compress the same dataset with a minimal set of parameters must be able to extract useful information from this data, which is what we commonly refer to as "generalization". Since we ultimately want to measure how well a model generalizes, not how well it stores information, we also measure the compression rate that accounts for the model size. In addition, compression rates that do not consider the model size might wrongly suggest that LLMs are preferable over general-purpose compressors, like gzip, to compress arbitrary files. Taking into account the model size shows that LLMs compression abilities come at the cost of "a complex compression algorithm".
>
>
> **Can you add a concrete example of how arithmetic coding works?**
>
> Yes, we visualize arithmetic encoding in Figure 1. Arithmetic coding iteratively partitions the interval $I = [0, 1)$ according to a predictive model $P$ and an input string, i.e., $AIXI$ for Figure 1.
>
> First, we construct the intervals for $P(\cdot)$:
>
> * $[0, 0.45)$ for $P(A) = 0.45$
> * $[0.45, 0.75)$ for $P(I) = 0.3$
> * $[0.75, 1)$ for $P(X) = 0.25$
>
> Since the first token is $A$, we set $I = [0, 0.45)$ and iterate. Thus, the intervals for $P(\cdot | A)$ are:
>
> * $[0.2 * 0, 0.2 * 0.45) = [0, 0.09)$ for $P(A | A) = 0.2$
> * $[0.09, (0.2 + 0.6) * 0.45) = [0.09, 0.36)$ for $P(I | A) = 0.3$
> * $[0.36, (0.2 + 0.6 + 0.2) * 0.45) = [0.36, 0.45)$ for $P(X | A) = 0.2$
>
> Since the next token is $I$, we set $I = [0, 0.45)$ and so on. We terminate with $I = [0.322, 0341)$ for $AIXI$. Next, arithmetic encoding will compute the binary sequence corresponding to iteratively splitting the interval $[0, 1)$ in half until it is fully contained in $I$.
>
> Concretely, this yields the binary sequence.
>
> * $b0$ -> $[0, 0.5)$
> * $b01$ -> $[0.25, 0.5)$
> * $b010$ -> $[0.25, 0.375)$
> * $b0101$ -> $[0.3125, 0.375)$
> * $b01010$ -> $[0.3125, 0.34375)$
> * $b010101$ -> $[0.328125, 0.34375)$
> * $b0101010$ -> $[0.328125, 0.3359375)$
>
> As $[0.328125, 0.3359375)$ is fully contained in $I = [0.322, 0341)$, the compressed output is $0101010$, which consists of 7 bits as opposed to the 4 bytes used to encode $AIXI$.
>
> We have extended the caption of the figure to clarify arithmetic coding further and included the above example in Appendix A.

---

> > ### Comment · Reviewer_hCy6 · 2023-11-22
> >
> > Thank you for addressing my concerns, and for clarifying my questions.
> >
> > As my major concerns have been addressed, it is only fair from my side to raise my score.
> >
> > However, I want to point out that while this work offers an intriguing perspective on large language models by examining them from a compression standpoint, the insights it provides, in my opinion, offers limited new understanding compared to what is already known from the "prediction perspective".

---

### Author Response · Authors · 2023-11-21
**General Response**

We thank the reviewers for their insightful comments and helpful feedback, which allowed us to significantly improve our submission. We are pleased that they emphasize "the novelty of applying LLMs to compressed coding of images and audio" (`R-hCy6`, `R-8SR8`, `R-NHbN`, `R-55Jb`), that they consider our "paper well-written" (`R-NHbN`) and "our experiments, tables, and figures easy to follow" (`R-hCy6`), and that they think "the paper should be presented at ICLR" (`R-55Jb`).

Here, we summarize our response to the common questions and we respond to the individual questions below each review.
* **Motivation**: We clarified the motivation for our paper, which is to popularize the connection between prediction and compression in language modeling and to demonstrate that foundation models are effective compressors (in particular on unseen data modalities).
* **Generative Modeling**: To show that "language modeling is compression", we first demonstrated that language models are strong compressors (Section 3.2), but we also needed to show that compressors give rise to auto-regressive sequence models (which is the motivation for Section 3.4). We extended the experimental results of Section 3.4 by including the generations of all Chinchilla models (i.e., 1B, 7B, and 70B) to visualize that a better compressor gives rise to higher-quality generations.
* **Arithmetic Coding and Figure 1**: We clarified the caption of Figure 1 and added a step-by-step example of arithmetic encoding to Appendix A.

We have also added the following requested experiments:
* **LLaMA Results**: We evaluated the compression capabilities of a model that is open-source, i.e., LLaMA1-7B, with arithmetic coding (see Table 1.) and showed that it performs comparably to the Chinchilla.
* **Open-Source Code**: We open-sourced our code and will link it to the paper upon acceptance.

---

### Meta-Review · Area_Chair_17rh · 2023-12-05

**Metareview:**

The paper investigates the hypothesis that language models compress data, and demonstrates that language models are strong compressors.

The paper received four reviews, all supportive of publication after the rebuttal. While the hypothesis that LLMs compress data is well known, the paper provides new and insightful experiments for this hypothesis. The reviewers emphasized that the paper provides useful insights, is well written and easy to understand, and adequately discusses limitations. The reviewers also voiced some concerns, like the perspective of language models performing compression being well known (55Jb), but after the rebuttal the reviewers agree that the insights and strength outweigh the weaknesses.

I also find the experiments useful as a base for the discussion whether a fundamental working principle of LLMs is compression.

**Justification For Why Not Higher Score:**

The hypothesis the paper investigates is widely known, and prior work already provides support for the hypothesis (e.g., Ge et al. (2023)). Therefore, while the paper is interesting for a presentation at the conference, additional value of the paper is not substantial enough for a spotlight or oral talk.

**Justification For Why Not Lower Score:**

The paper provides needed evidence for the interesting hypothesis that a working principe of LLMs is compression.

---

### Decision · Program_Chairs · 2024-01-16

Accept (poster)